# Named Entity Recognition Model Based on Feature Fusion

**Zhen Sun \* and Xinfu Li**

School of Cyberspace Security and Computer Science, Hebei University, Baoding 071000, China
* Correspondence: 20191366@stumail.hbu.edu.cn; Tel.: +86-178-0037-6608

**Abstract:** Named entity recognition can deeply explore semantic features and enhance the ability of vector representation of text data. This paper proposes a named entity recognition method based on multi-head attention to aim at the problem of fuzzy lexical boundary in Chinese named entity recognition. Firstly, Word2vec is used to extract word vectors, HMM is used to extract boundary vectors, ALBERT is used to extract character vectors, the Feedforward-attention mechanism is used to fuse the three vectors, and then the fused vectors representation is used to remove features by BiLSTM. Then multi-head attention is used to mine the potential word information in the text features. Finally, the text label classification results are output after the conditional random field screening. Through the verification of WeiboNER, MSRA, and CLUENER2020 datasets, the results show that the proposed algorithm can effectively improve the performance of named entity recognition.

**Keywords:** named entity recognition; ALBERT; vector fusion; multiple head attention

## 1. Introduction

Named Entity Recognition (NER) is an essential task in natural language processing [1]. Combining computer science and linguistics, NER studies various theories and methods for effective communication between humans and computers using natural language, aiming to extract specific entities from unstructured text relationships [2]. For example, names of people, places, organizations [3], etc.

In machine learning methods, NER is usually treated as a sequence annotation task [4].

The neural network model usually includes three parts: the embedding layer, the encoding layer and the output layer [5]. The marker model is learned by large-scale corpus, and the word information is annotated by combining the word location information and vector representation. In the embedded layer, the pre-trained model is mainly used to learn the distributed representation of text, and Bert [6] uses MLM and SOP tasks to achieve good results.

The second part is the encoding layer, which is used to extract the sequence features and then capture the context dependencies of the input text features. Using CNN for coding has high parallel computing efficiency but weak feature extraction ability on long sequence input. BiLSTM [7] is proposed to carry out feature coding to learn context dependencies.

The third part is the output layer, which extracts the encoding of the second part, generates the optimal tag sequence, and obtains the tag recognition result of the NER task. Softmax function is widely used in multiple classification tasks, and often ignores label interdependence in sequence labeling tasks. Therefore, CRF is mainly used to learn the label dependence of named entities [8] and becomes the first choice of the NER task decoding layer.

Compared with English-named entities, Chinese-named entities have no obvious word boundary [9], which significantly affects the accuracy of Chinese-named entity recognition. However, Chinese characters have different meanings in different scenarios, but the previous research on Chinese-named entity recognition only transferred the named entity

recognition method in the English field to Chinese-named entity recognition, without considering the meaning of the word in the specific context, the word level characteristics, and the influence brought by the entity boundary. which makes the network model unable to fully extract the potential semantic relations of words [10]. Characters or words represent as vectors. Although rich semantic features of the text are obtained, the understanding of the text is not sufficient, and the word sense information is not integrated. Although the simple fusion of word vectors and word vectors can improve the accuracy of the named entity recognition task, it cannot fully mine the word boundary information and pay attention to the critical features in the text.

Aiming at the above problems in the Chinese named entity recognition task, this paper proposes a named entity recognition model combining multi-granularity fusion and multi-head attention. The main contributions are as follows: Integrating boundary and character features into named entity recognition models, The attention mechanism combines character vector, boundary vector, and word vector into BiLSTM to extract features. More weights are assigned to critical elements, and the influence of incorrect boundary division of named entity recognition vocabulary is reduced. At the same time, relatively rich semantic features of the Chinese language are obtained. The multi-head attention mechanism is used to adjust the output weight of BiLSTM, capture the multiple semantic features, and fully reflect the close relationship between each word.

## 2. Related Works

Named entity recognition is an essential primary tool in information extraction [11], machine translation [12], question-answering systems [13], syntactic analysis [14], and other application fields. It is essential in natural language processing technology's practical application [15,16]. At present, named entity recognition is mainly divided into rule and dictionary-based methods, traditional machine learning-based methods, and recently popular neural network-based methods.

Most methods based on rules and dictionaries need to construct rule templates manually, select specific plans, including statistical data, indicator words, position words, centre words, etc., complete named entity recognition by pattern and string matching, and assign corresponding weights to each rule. When a rule conflict occurs, select the law with the most significant value to determine the named entity type. Generally speaking, when the extracted rules can accurately reflect the language and phenomenon, rule-based methods are usually chosen. For example, Collins [17] et al. proposed the DL-Coltrain plan, and Rau [18] et al. proposed manual rule writing and a heuristic algorithm. These methods are generally based on establishing a knowledge base and dictionary and need to assign a certain weight to each practice. Rules are often related to specific languages and domains. Manually specifying rules is time-consuming and error-prone, and difficult to migrate. Therefore, many researchers began to study machine learning methods in the field of NER.

Methods based on traditional machine learning include Traditional machine learning methods such as the hidden Markov model [19], Conditional Random Field (CRF) [20], and support vector machine [21], which rely on artificial feature extraction. The final model effect also depends on the advantages and disadvantages of artificial feature extraction. With the wide application of deep learning in image processing, computer vision, and other aspects, researchers have begun to realize the importance of deep knowledge in natural language processing, and the deep learning model that takes named entity recognition as a sequence annotation task has gradually emerged in the field of natural language processing.

Unlike English named entity recognition tasks, Chinese words do not have obvious lexical boundaries, so the accuracy of Chinese word segmentation will directly impact the effect of named entity recognition [22]. In the current work of Chinese-named entity recognition, Chinese-named entity recognition is regarded as a sequence annotation task. Unanue et al. [23] proposed a research method of named entity recognition based

on RNN-CRF. This model does not rely on artificial features, uses neural networks to extract local information and vectorize the representation of text, and combines text context information for named entity recognition. Collobert et al. [24] proposed using CNN to capture text structure information through a fixed-length window near a given location. They achieved good results in named entity recognition based on lexical features. In recent years, an attention-mechanized Transformer structure has become popular in the natural language processing space due to its ability to capture context at a distance and parallelism. However, the attention-mechanized Transformer [12] structure is far less effective than the LSTM network when it comes to named entity recognition tasks.

Lexis-based NER cannot obtain the entity boundary well, and the division of linguistic information depends more on the accuracy of Chinese word segmentation. Chen et al. [25] proposed using Chinese characters to enhance the effect of Chinese word vector training and integrated character embedding into word embedding in the way of multi-prototype character embedding. However, this method needs to identify Chinese characters in advance and build a word list, and the richness of the word list limits the recognition accuracy of named entities. Zhang et al. [26] proposed the Lattice model of integrating lexical information into character information, which has played a good role in enhancing linguistic knowledge. However, the computational performance is relatively low, and the operation cannot be parallelized. Moreover, each character cannot obtain verbal communication in front of it, which is easy to cause information loss.

Chinese named entity recognition task facing the deep learning method is one of the difficulties of solving the problem of text words fuzzy boundaries, previously called entity recognition in the use of the form of word vectors often leads to difficulties of word segmentation. The emergence of the character vector based on word vector segmentation, to some extent, avoids the cause of the polysemy phenomenon [27,28]. As well, the character vector is far less than the word vector, so using the word-level vector in the text vector can reduce the difficulty of solving the above problem. However, a single character vector may lead to the loss of information in the text, and the lack of lexical boundaries will make the model unable to fully understand the semantic information of the text. Therefore, we use the fusion of lexical vector and word vector to improve the model's semantic representation ability of Chinese text, and adjust the weight relationship of different vectors through the attention mechanism to enhance the understanding of key semantic information in the text.

## 3. Our Methodology

This paper proposes a named entity recognition model based on multi-head attention. The model is divided into four layers: embedding, encoding, attention, and decoding. The model embedding layer obtains the vector representation of the text, the encoding layer extracts the vector representation for other features, the attention layer strengthens the semantic relationship of the text based on extracting features, and the output layer does the final label decoding.

The overall structure of the model is shown in Figure 1:

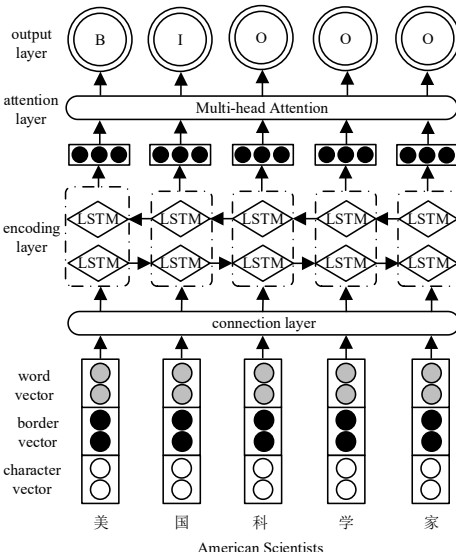

**Figure 1.** Model framework.

The model mainly uses Word2vec to obtain word vectors, HMM to obtain boundary vectors, ALBERT to obtain dynamic word vectors, and attention mechanism to allocate the weight of vectors in the text. The multi-granularity vector representation is transferred to the BiLSTM network layer for vector feature extraction. Then the output of the hidden layer is combined with the self-attention mechanism. Multi-head attention is used to learn the dependence relationship between words and capture the deep information of word vectors. Finally, the corresponding relationship between labels is found by CRF, the whole hidden state is modeled, and the label information is screened according to the weight allocation of titles.

### 3.1. Embedded Layer

In the named entity recognition task, obtaining high-quality vector representation is the key to text semantic understanding. Although word vectors are rich in semantics, they are limited by Chinese word segmentation, and some errors in word segmentation may affect the results of named entity recognition. For example, the accurate entity label of the word "南京市长江大桥 (Nanjing Yangtze River Bridge)" is "position", jieba word segmentation tool divides it into two entities, "南京市长 (Nanjing Mayor)" and "江大桥 (Jiang Bridge)", which are respectively marked with "position" and "name", thus generating incorrect identification results. However, there is a gap between the semantic expression of a single Chinese character and the semantic face of the text. The combination of character and word vectors can overcome the impact of word segmentation results on the task of named entity recognition. At the same time, the attention mechanism can adjust the weight of the word vector and focus on the vital information in the text semantics.

#### 3.1.1. Enhanced Word Vectors

(1)  Extract word vectors

In data processing, one-hot coding leads to high data sparsity and large dimensions, which may bring dimensional disaster in a large corpus, making it challenging to reflect the relationship between words. However, Word2Vec does not need large-scale manual labeling samples and can effectively reduce the vector dimension. It can express the relationship between words by calculating the cosine similarity between vectors. The word vector is obtained for a given input sequence $\{C_1, C_2, \ldots, C_n\}$ by pre-training the Word2Vec model as $x_i^w = \{x_1, x_2, \ldots, x_n\}$.

(2)  Extract word boundary vectors

The word vector only considers the meaning and the word itself and does not consider the boundary information between words. The boundary information in the text takes into account the position of characters in word entities, which is beneficial to enhance the understanding of semantic information. In this paper, for a given sentence X, the jieba word segmentation tool and HMM are used to segment entities. The entity is labelled with the BIO tag method. The character at the beginning position of the entity is marked as "B". Characters in the middle and end of the entity are marked with "I"; Characters that are not entity types are marked as "O". For a given input sequence $\{C_1, C_2, \ldots, C_n\}$, the word vector obtained by HMM is $x_i^b = \{x_1, x_2, \ldots, x_n\}$.

Finally, the enhanced word vector representation $x_i = x_i^w + x_i^b$ can be obtained.

### 3.1.2. Albert Extracts Word Vectors

ALBERT [29] is a lightweight pre-training model developed based on the BERT pre-training model. ALBERT has a similar performance to BERT through parameter factorization, cross-layer parameter sharing, and inter-sentence order prediction, which effectively reduces the number of parameters. The model architecture is shown in Figure 2:

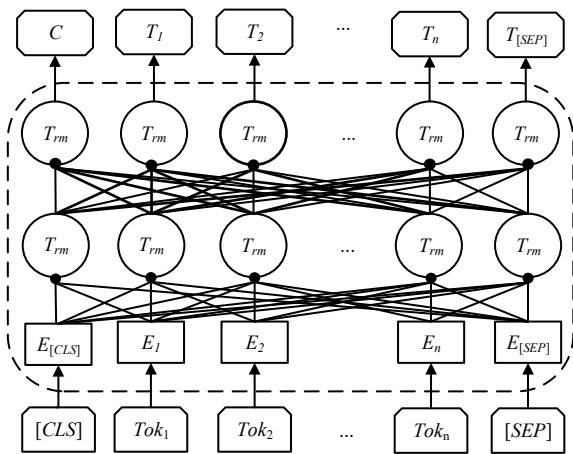

**Figure 2.** ALBERT model.

Where $E_i$ and $T_i$ respectively represent the vector representation corresponding to the text sequence of the i word in the text data and the word vector obtained after Transformer encoding, $T_{rm}$ represents the two-way Transformer module inside ALBERT, with a total of 12 layers. Let the input statement S $= c_1 c_2 \ldots c_t$, where $c_i \in C, i = 1, 2 \ldots t$. The word vector $X = [x_{c_1}; x_{c_2}; \ldots; x_{c_t}]$ corresponding to $y_i = \{y_1, y_2, \ldots, y_n\}$ is obtained by ALBERT.

### 3.1.3. Vector Fusion

The attention mechanism can dynamically adjust the proportion of character vectors and enhanced word vectors, which can better modify the text semantics represented by different vectors. The attention mechanism can effectively reduce the vector's dimension and reduce the model's complexity.

Considering the influence of different vectors, character vectors and word vectors were fused through the feedforward-attention mechanism [30]. The attention mechanism is divided into two steps: first, the attention weight distribution of all input vectors is calculated; Secondly, the weight distribution of attention is used to assign weights to different vectors.

The attention score corresponding to the word vector is shown in Equation (1):

$$p(x_i) = \sigma\left(W_2^T \tanh\left(W_1^T x_i + b_1\right) + b_2\right) \tag{1}$$

The attention score corresponding to the character vector is shown in Equation (2):

$$p(y_i) = \sigma\left(W_2^T \tanh\left(W_1^T y_i + b_1\right) + b_2\right) \tag{2}$$

Text vector output can be obtained after the fusion of the two, as shown in Equation (3):

$$c_i = \sum_{i=1}^{n} (p(x_i) \times x_i + p(y_i) \times y_i) \tag{3}$$

Among them, $w_1, w_2 \, \epsilon \, R^{1*dim}, b_1, b_2 \, \epsilon \, R$ are all learnable weight coefficients. The corresponding fusion mechanism is shown in Figure 3:

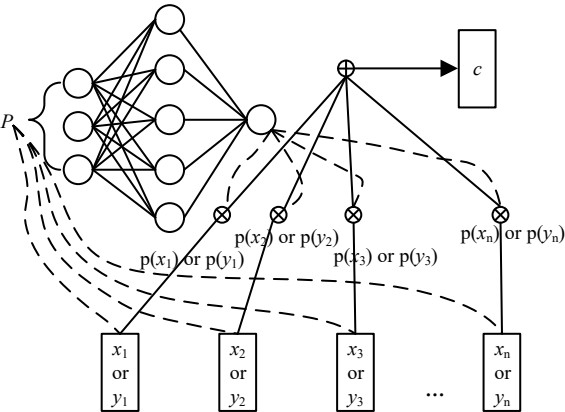

**Figure 3.** Fusion mechanism.

Different from the simple splicing method, the Feedforward-attention mechanism focuses on the internal relations of text vectors based on the original vector representation, improves the sensitivity of the coding layer to input vectors by using weighted summation, and makes full use of the contextual semantic information of the text.

### 3.2. Coding Layer

LSTM is a very suitable RNN algorithm for the modeling of text lexical features, which can solve the problems of gradient disappearance and gradient explosion to a certain extent. The LSTM model is shown in Figure 4:

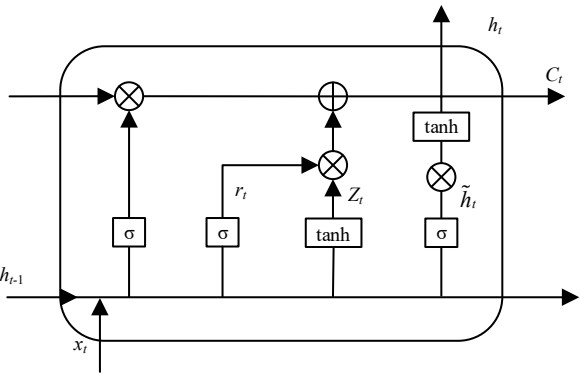

**Figure 4.** LSTM model.

However, LSTM cannot encode words from front to back and excavate more sequence feature vectors. BiLSTM obtains the forward and backward vectors of the sequence through the two hidden layers of the forward and backward LSTM, respectively, and then fuses the two vectors to obtain the new feature representation of the text sequence.

Let $C = [C_1, C_2, \ldots, C_T]$ be the input vector representation of BiLSTM, and the implementation of the internal structure of LSTM neurons is shown in Equations (4)–(8):

$$i_t = \sigma\big(w_i[w_{h_i}, c_t] + b_i\big) \tag{4}$$

$$f_t = \sigma\Big(w_f[h_{t-1}, c_t] + b_i\Big) \tag{5}$$

$$n_t = \tanh(w_n[h_{t-1}, c_t] + b_n) \tag{6}$$

$$o_t = \sigma(w_{x_o}[h_{t-1}, c_t] + b_o) \tag{7}$$

$$h_t = o_t \tan h(n_t) \tag{8}$$

where: $i_t$, $h_t$, $f_t$, $n_t$, $o_t$ are the states of the memory gate, hidden layer, forgotten gate, nucleus, and output gate when the $t$ text command is input. $w$ is the model parameter, $b$ is the bias vector, $\sigma$ is the sigmoid function, and tanh is the hyperbolic tangent function.

### 3.3. Multiple Layers of Attention

Although BiLSTM can extract the long-distance dependence relationship of text, the characteristics of timing make it difficult to extract local information and fail to highlight the key information. The self-attention mechanism focuses on the correlation of internal features, reduces the dependence on external information, enables the whole model to better obtain the dependence of words, and effectively utilizes the latent semantic information. Based on the self-attention mechanism, multi-layer self-attention is used, and the number of self-attention layers is a hyperparameter. It is better to attach different degrees of weight to different feature vectors in text vectors, extract key local features, and generate a joint feature vector matrix containing global features and local features.

For example, in the sentence " 亚马逊森林是珍贵的自然资源宝库 (The Amazon forest is a treasure trove of precious natural resources)", the relationship between the words of the named entity " 亚马逊森林 (The Amazon forest)" is closer, and more weight will be allocated to the attention mechanism. The correlation degree of other words in the sentence is weak, and the weight assigned is small. At the same time, " 自然资源宝库 (Treasure house of natural resources)" plays a positive role in accurately identifying the named entity " 亚马逊森林 (The Amazon forest)", so more weight will be assigned to " 自然资源宝库 (Treasure house of natural resources)".

To fully capture the dependencies in the long range of sentences and the characteristics of comprehensive sentences, we introduce the multi-head attention mechanism into the network model. Multi-head attention is an improvement of self-attention, which reduces the dependence on external attention and focuses on the internal association of capturing text features, which is represented as the stacking of multiple self-attention layers in the structure. The self-attention function itself can be expressed as the mapping between Query and key value correlation. The calculation of attention is divided into three steps. The first step is to calculate the similarity between the Query and key value to obtain the corresponding weight. The second step is to use the softmax function to normalize the weight. Finally, the weight and the related key value are calculated to obtain the corresponding probability distribution of attention. As a special case of attention mechanism, self-attention exists $Q = K = V$. Units in each sequence and all units in that sequence can be counted for attention. Let the input sequence be $C = \{c_i\}_1^N$, and the output of the text vector obtained by BiLSTM be $B_i = \{B_i\}_1^N$. Then multi-granularity embedded feature vectors are processed by multi-head attention. Firstly, the parameter matrix is used to map the current feature vector h times, as shown in Equations (9)–(11):

$$Q = W_q B_i \epsilon R^{n \times dk} \tag{9}$$

$$K = W_k B_i \epsilon R^{m \times dk} \tag{10}$$

$$V = W_v B_i \in R^{m \times dk} \tag{11}$$

where $d\_k$ represents the second dimension of $Q$ and $K$, the scaled dot product operation is carried out on $Q$, $K$, and $V$ each time. The scaled dot product attention is shown in Equation (12):

$$Attention(Q, K, V) = softmax\left(\frac{QK^T}{d_k}\right)V \tag{12}$$

where $d_k$ is a dimension of $Q$ and $K$, which is positively related to the size of the inner product of $Q$ and $K$, such that $Q = K = V$, softmax normalizes in the M dimension. Multi-head attention is the encoding of $K$, $Q$, and $V$ of feature vectors at different positions, get $Q_i = QW_i^Q$, $K_i = KW_i^K$, $V_i = VW_i^V$ through different projection matrices $W_i^Q, W_i^K$, $W_i^V \epsilon R^{d \times d'}$, and then do Attention, as shown in Equation (13):

$$MultiHead(Q, K, V) = Concat(head_1, \ldots, head_n)W^o \tag{13}$$

$$where\ head_i = Attention\left(QW_i^Q, KW_i^K, VW_i^V\right)$$

Among them, $W_i^Q \epsilon \mathbb{R}^{d_{model} \times d_k}$, $W_i^K \epsilon \mathbb{R}^{d_{model} \times d_k}$, $W_i^V \epsilon \mathbb{R}^{d_{model} \times d_k}$, $W^o \epsilon \mathbb{R}^{d_{model} \times d_v}$ The structure of the multi-head attention mechanism is shown in Figure 5:

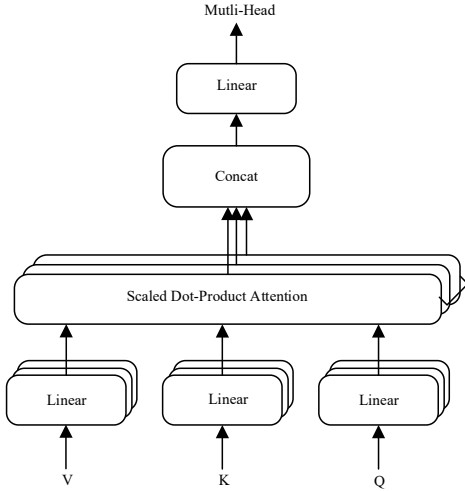

**Figure 5.** Multi-head attention.

According to Formula (13), the output of the multi-head Attention layer can be obtained. Combined with the calculation process in Figure 5, $d_q = d_k = d_v$ can also be obtained.

### 3.4. Output Layer

The discriminative method of a bidirectional long short-term memory network ignores the correlation of output category labels. Conditional random fields can be used to further constrain labels according to the sequence scores obtained by the long short-term memory network. For a given input sequence $\{C_1, C_2, \ldots, C_n\}$, the corresponding accurate label sequence is $\{S_1, S_2, \ldots, S_n\}$, then the linear random field of the chain element can be obtained.

$$P(S_i | C, S_1, S_2, \ldots, S_{n-1}, S_n) = P(S_i | C, S_{n-1}, S_{n+1}) \tag{14}$$

In this case, $P(S|C)$ can be used to represent the probability distribution of random fields of linear chain elements:

$$P(S|C) = \frac{1}{Z(c)} \exp\left\{\sum_{i,k} \lambda_k, f_k(s_{i-1}, s_i, c, i) + \sum_{i,l} \mu_l h_l(s_i, c, i)\right\} \tag{15}$$

where $Z(C) = \sum_s exp\{\sum_{i,k} \lambda_k f_k(s_{i-1}, s_i, c, i)\}$ represents the normalization factor, *f, h* represents the characteristic function, and $\lambda_k, \mu_1$ are the corresponding weight coefficient of the characteristic function.

According to the given text sequence *C* and the label sequence with the maximum probability obtained, the output sequence $\hat{S}$ can be obtained, as shown in Equation (16):

$$\hat{S} = argmaxP(S|C) \tag{16}$$

## 4. Results and Discussion

### 4.1. The Data Set

Three datasets of WeiboNER, Microsoft MSRA, and CLUENER2020 are experimental datasets of the model. Public data sets have been used and verified in many articles. Different entity types are used to annotate data types. Table 1 describes these three data sets.

**Table 1.** Experiment data set.

| Data Set | Type | The Training Set | The Test Set | Validation Set |
|---|---|---|---|---|
| WeiboNER | sentence | 1.4 k | 0.3 k | 0.3 k |
| | character | 73.5 k | 14.4 k | 14.8 k |
| Microsoft MSRA | sentence | 46.1 k | 4.4 k | - |
| | character | 2169.9 k | 4.4 k | |
| Cluener2020 | sentence | 10.7 k | 13.4 k | 13.4 k |
| | character | 41.3 k | 13.4 k | 51.6 k |

### 4.2. Data Set Annotation and Evaluation Criteria

The annotation format is the BIO format. For example, B-org, I-org, or O are used for the organization name. The evaluation criteria adopted are the same as those of previous relevant papers, namely, P (Precision), R (Recall), and F1 values are used to evaluate the effect of the model. The calculation process of each evaluation index is shown in Equations (17)–(19):

$$P = \frac{TP}{TP + FP} \tag{17}$$

$$R = \frac{TP}{TP + FN} \tag{18}$$

$$F1\text{-}score = \frac{2PR}{P + R} \tag{19}$$

*TP* is to predict the positive class as the positive class, *FP* is to predict the negative class as the positive class, and *FN* is to predict the original positive class as the negative class. *F1-score* takes into account both the precision and recall of the classification model and can be regarded as a weighted average of precision and recall.

### 4.3. Comparative Experiment and Result Analysis

#### 4.3.1. Comparative Experiment

To verify the effectiveness of the model proposed in this paper, the effectiveness of the model is compared with several models listed below. The specific situation of each model is as follows:

Lattice-LSTM [26]: Lattice-LSTM model is an improved, enhanced semantic vector representation model based on the LSTM model. It uses lattice structure LSTM to represent words in the text. It integrates potential word vectors into the LSTM network based on word vectors, which has achieved good results in Chinese-named entity recognition experiments.

Lr-CNN [2]: Aiming at the defect that Lattice-LSTM cannot operate in parallel, LR-CNN proposes that CNN is used to encode word vectors, and different receptive fields are

used to extract features. Meanwhile, the feedback layer is combined with adjusting the weight of word vectors.

CGN [31]: The word vector is integrated into each word vector in the vocabulary. The word vector is fused into the upper and lower word vectors using the word boundary vector and the GAN graph attention network.

FLAT [32]: The location information of the transformer model is improved, and different encodings of characters and vocabulary are constructed, respectively. As an encoder, Transformer is more suitable for Chinese-named entity recognition.

Glyce [33]: Proposed a CNN-based hieroglyphic recognition Tianzige-CNN architecture, which combined the glyph vector with the traditional word vector and added the image classification loss function to optimize for the characteristics of Chinese-named entity recognition.

The proposed model compares several mainstream lexical enhancement methods for named entity recognition. The experimental results shown in Table 2 show that, the proposed model is significantly better than other models on the WeiboNER dataset, which proves the advantages of the proposed word-word fusion model in a named entity recognition task. The experimental results on the MSRA dataset were second only to Glyce. This is because the Glyce model combines character information and glyph information to provide better text information when processing large-scale corpora, and it works well on MSRA datasets. WeiboNER corpora has smaller data sets, including more social media comments on information, the text data is mixed and disorderly, Glyce in the characterization of such characteristics of text is still lacking, in multiple feature fusion under the said it is difficult to identify the characteristics of a messy text data, and the proposed model in small training sample data set has a very good effect, Therefore, the model presented in this paper has good robustness against datasets with different text quantities. Lattice-lstm, LR-CNN, CGN, and FLAT models are based on two feature representations: character and lexical vector. The multi-head attention is combined based on the fusion of word and lexical vector representation in this model, which makes the model pay more attention to the potential semantic features of text and makes up for the defects of traditional neural network BiLSTM in feature extraction. The classifier can achieve a better effect on entity recognition. Compared to other fusion vocabulary enhancement of text vector method, this paper model considers both the vector fusion, based on the adaptive coding to reduce the error word boundaries, and USES the long attention mechanism further attention to text context dependencies, proved in this paper, the model in the role of the Chinese named entity recognition task.

**Table 2.** Comparison of named entity recognition results of different models.

| Model | MSRA | | | WeiboNER | | |
|---|---|---|---|---|---|---|
| | **P** | **R** | **F1** | **P** | **R** | **F1** |
| Lattice-LSTM | 93.57% | 92.79% | 93.18% | - | - | 58.79% |
| LR-CNN | 94.50% | 92.93% | 93.71% | - | - | 59.92% |
| CGN | 94.01% | 92.93% | 93.47% | - | - | 63.09% |
| FLAT | - | - | 94.35% | - | - | 67.71% |
| Glyce | 95.57% | **95.51%** | **95.54%** | 67.78% | 67.71% | 67.60% |
| Our model | **95.81%** | 91.97% | 94.78% | **72.57%** | **70.31%** | **71.42%** |

### 4.3.2. Ablation Experiment

To further verify the effectiveness of the model proposed in this paper, BILSTM-CRF model and Albert-BiLSTM-CRF model were established respectively in this paper. On this basis, the Feedforward-attention mechanism was used to conduct control experiments on three kinds of data sets with the intermediate model formed by the three kinds of vectors. To verify the effect of the fusion vector module and multi-head attention module on the model.

From the experimental results in Table 3 show that, we can see that the feature representation of word and word vector fusion and the named entity recognition effect of fusion vector + BILSTM-CRF model are improved on the three types of data sets, which proves the effectiveness of word and word fusion method and Feedforward-attention fusion mechanism. Compared with the direct vector splicing method, this paper uses feedforward attention to assign the weight of vector representation and pays more attention to the information that may become words in sentences, which verifies the effectiveness of the submodule of word and word information fusion in this paper.

**Table 3.** Ablation results.

| Model | CLUENER2020 | WeiboNER | MSRA |
|---|---|---|---|
| BiLSTM-CRF | 70.41% | 58.76% | 83.41% |
| ALBERT-BiLSTM-CRF | 79.63% | 68.59% | 92.61% |
| Stitching vector | 80.32% | 68.75% | 93.52% |
| Fusion vector | 81.59% | 70.35% | 93.57% |
| Our model | **82.47%** | **71.42%** | **94.78%** |

In comparison, the model integrating word and word vectors does not improve much on the MSRA dataset but greatly improves on CLUENER2020 and WeiboNER datasets. This is because the vector representation method integrating word vectors has limited improvement in entity boundary division on large regular datasets. For the relatively scattered Internet datasets WeiboNER and CLUENER2020, the fusion of character vector and vocabulary vector have a better improvement effect. It can be seen that the fusion vector effectively solves the difficult problem of entity boundary recognition by static word vectors, which is helpful to enhance the representation of lexical information and improve the effect of named entity recognition.

Compared with the model proposed in this paper, the F1 value of CLUENER2020, WeiboNER, and MSRA on the three datasets can be improved by 0.88%, 1.07%, and 1.21%. The F1 value of the model using multi-head attention can be significantly improved. This shows that the model incorporating multi-head attention improves the BiLSTM encoding layer's processing of text dependence, further improves the model's effect of extracting context dependence, and verifies the effectiveness of the multi-head attention sub-module.

To verify the effectiveness of using multi-head attention to extract contextual features, this paper also uses different levels of attention for analysis. The experimental results on the CLUENER2020 dataset are shown in Figure 6. Among them, the use of 0 layers of the self-attention layer means the unused attention layer, and the use of 1 layer of the self-attention layer means the fusion model only combines the self-attention.

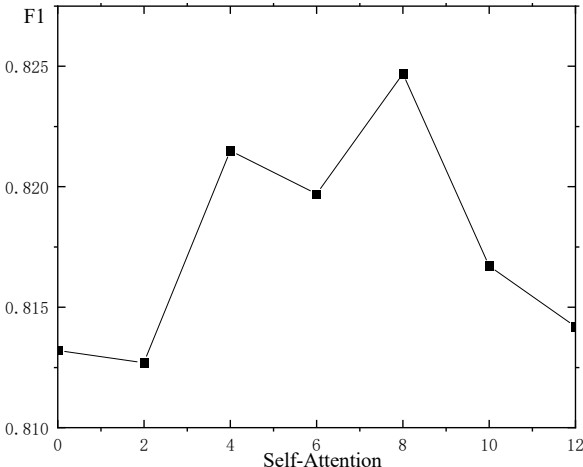

**Figure 6.** Experimental results with different levels of self-attention.

From the experimental results, it can be seen that the effect of the model is significantly improved after adding the self-attention mechanism. As well, the best effect was achieved when the number of self-attention layers reached 8, focusing on the local dependence of the text. When the number of layers exceeds 8, multihead attention will focus on redundant linguistic information, reducing the effect of model learning.

## 5. Conclusions

In this paper, Word2vec is used to obtain word vectors, HMM is used to obtain boundary vectors, the ALBERT pre-training model is used to encode Chinese text character vectors, and three kinds of vectors are fused into BiLSTM neural network through attention mechanism to obtain semantic features of text statements. Then the multi-head attention mechanism is used to capture the correlation information between the characters in the statement. Finally, CRF decoding is used to construct a new Chinese-named entity recognition model.

The experimental results show that the model is better than the mainstream model. The relationship between characters and the semantic division of Chinese words makes vector fusion very important. Multi-head attention can strengthen the internal relationship between texts and find more effective information between vectors, which has a unique advantage in Chinese named entity recognition. In future work, the proposed model will be transferred to other fields of named entity recognition research to lay a foundation for machine translation, relation extraction, and other tasks.

**Author Contributions:** Investigation, Z.S.; Methodology, Z.S.; Software, X.L.; Writing—original draft, Z.S.; Writing—review & editing, X.L. All authors have read and agreed to the published version of the manuscript.

**Funding:** This research received no external funding.

**Institutional Review Board Statement:** Not applicable.

**Informed Consent Statement:** Not applicable.

**Data Availability Statement:** The raw data MSRA, WeiboNER, CLUENER2020 required to reproduce the above findings are available to download from https://github.com/OYE93/Chinese-NLP-Corpus/tree/master/NER/MSRA (accessed on 1 August 2020), https://github.com/OYE93/Chinese-NLP-Corpus/tree/master/NER/Weibo (accessed on 1 August 2020), https://github.com/CLUEbenchmark/CLUENER2020 (accessed on 1 August 2020).

**Conflicts of Interest:** The authors declare no conflict of interest.

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
