# Peer review of "Named Entity Recognition Model Based on Feature Fusion"

_information, doi:10.3390/info14020133_

Round 1
Reviewer 1 Report
In this manuscript, the authors propose work on NER by proposing a multi-headed attention-based named entity recognition method to address the problem of fuzzy lexical boundaries in the Chinese language named entity recognition.
W2V is used to extract vectors of words, an HMM is used to extract boundary vectors, ALBERT is used to extract character vectors, the Feedforward-attention mechanism is used to merge the three vectors, and then the representation of merged vectors is used to remove features by BiLSTM. Furthermore, multi-headed attention is used to extract information about potential words in text features. Finally, text label classification results are output after conditional random field screening. Through the verification of WeiboNER, MSRA, and CLUENER2020 datasets, the results show that the proposed algorithm can effectively improve the performance of named entity recognition.
The introduction is not well defined. In fact, it is not clear what new and unexplored things the authors are bringing. The authors should emphasize their innovation.
The methodology section (Section 3) seems to collect all architectures that have been seen before. I recommend that the authors add a dash of innovation and streamline what already exists in the state of the art. Make their contribution deeper at the descriptive level.
I would remove Table 2. While important, the info can be described at the end of the section. Same goes for Tab 3.
In the experiments table, I strongly recommend using a bold font to highlight performance.
Figure6 has a too-long explanation. I suggest making the description much more streamlined.
I recommend making your code available through GitHub to increase reader interest.
Finally, I formally recommend that you evaluate the symbolic meaning of transformers by including in the background a work that might help you (https://www.mdpi.com/1999-5903/14/1/10).
Author Response
This paper combs the description of the introduction part and restates the innovation points of the paper.
Add innovative description in the method section to simplify the existing description method.
Tables 2 and 3 were dropped.
Use bold to highlight performance.
The redundant description in Figure 6 has been removed.
After publication, the authors submit the code to github.
Redescribe transformer in the relevant working section.

Reviewer 2 Report
The article presents a neural network architecture for Named Entity Recognition composed of a BiLSTM followed by a multi-head attention. The output layer uses CRF to emit the label.
Since your using Multi-layer attention in your neural network, it could be interesting to substitute the BiLSTM in your model with a Transformer. Did you test it?
I cannot understand Figure 3. What do you mean with "x_1 or y_1" and the corresponding "a(x_i) or a(y_i)", i.e. what do you mean with the word "or"?
If your idea is to select either x_i or y_i, you can use an highway network to calculate a score p for x_i (and 1-p for y_i).
Minor revision:
- I would like to suggest to change the word "decoder" with "output layer" to distinguish the decoder architecture with the last layer of a neural network.
- again in section 2.1, add a reference for jieba word segmentation tool.
- Equation 2 in Section 2.1.3: substitute the x_i element with y_i
- Section 2.3 I would suggest to add the translation to the Chinese text for foreign readers. Same thing for the input of Figure 1.
- please, improve section 2.1; I found it very difficult to follow. Add some examples to explain your input and your output (where it is possible).
Author Response
According to the opinions, the author adds transformer as an encoder experiment. However, the experiment effect is not good. It will be discussed in the relevant work section based on the work of others.
The formula was revised again.
Change the "decoder" of the paper to "output layer".
Added the jieba segmentation tool in 2.1.
Corrected the formula.
Insert a translation into the paper.
Examples are added in the chapter.
